# Identifying through Flows
# for Recovering Latent Representations

**Shen Li**
Institute of Data Science & NUS Graduate School for Integrative Sciences and Engineering
National University of Singapore
`shen.li@u.nus.edu`

**Bryan Hooi & Gim Hee Lee**
Department of Computer Science, National University of Singapore
`{bhooi,gimhee.lee}@comp.nus.edu.sg`

## Abstract

Identifiability, or recovery of the true latent representations from which the observed data originates, is *de facto* a fundamental goal of representation learning. Yet, most deep generative models do not address the question of identifiability, and thus fail to deliver on the promise of the recovery of the true latent sources that generate the observations. Recent work proposed identifiable generative modelling using variational autoencoders (iVAE) with a theory of identifiability. Due to the intractablity of KL divergence between variational approximate posterior and the true posterior, however, iVAE has to maximize the evidence lower bound (ELBO) of the marginal likelihood, leading to suboptimal solutions in both theory and practice. In contrast, we propose an identifiable framework for estimating latent representations using a flow-based model (iFlow). Our approach directly maximizes the marginal likelihood, allowing for theoretical guarantees on identifiability, thereby dispensing with variational approximations. We derive its optimization objective in analytical form, making it possible to train iFlow in an end-to-end manner. Simulations on synthetic data validate the correctness and effectiveness of our proposed method and demonstrate its practical advantages over other existing methods.

## 1 Introduction

A fundamental question in representation learning relates to identifiability: under which condition is it possible to recover the true latent representations that generate the observed data? Most existing likelihood-based approaches for deep generative modelling, such as Variational Autoencoders (VAE) (Kingma & Welling, 2013) and flow-based models (Kobyzev et al., 2019), focus on performing latent-variable inference and efficient data synthesis, but do not address the question of identifiability, i.e. recovering the true latent representations.

The question of identifiability is closely related to the goal of learning *disentangled* representations (Bengio et al., 2013). While there is no canonical definition for this term, we adopt the one where individual latent units are sensitive to changes in single generative factors while being relatively invariant to nuisance factors (Bengio et al., 2013). A good representation for human faces, for example, should encompass different latent factors that separately encode different attributes including gender, hair color, facial expression, etc. By aiming to recover the true latent representation, identifiable models also allow for principled disentanglement; this suggests that rather than being entangled in disentanglement learning in a completely unsupervised manner, we go a step further towards identifiability, since existing literature on disentangled representation learning, such as $\beta$-VAE (Higgins et al., 2017), $\beta$-TCVAE (Chen et al., 2018), DIP-VAE (Kumar et al., 2017) and FactorVAE (Kim & Mnih, 2018), are neither general endeavors to achieve identifiability; nor do they provide theoretical guarantees on recovering the true latent sources.

Recently, Khemakhem et al. (2019) introduced a theory of identifiability for deep generative models, based upon which the authors proposed an identifiable variant of VAEs called iVAE, to learn the distribution over latent variables in an identifiable manner. However, the downside of learning such an identifiable model within the VAE framework lies in the intractability of KL divergence between the approximate posterior and the true posterior. Consequently, in both theory and practice, iVAE inevitably leads to a suboptimal solution, which renders the learned model far less identifiable.

In this paper, aiming at avoiding such a pitfall, we propose to learn an identifiable generative model through flows (short for normalizing flows (Tabak et al., 2010; Rezende & Mohamed, 2015)). A normalizing flow is a transformation of a simple probability distribution (e.g. a standard normal) into a more complex probability distribution by a composition of a series of invertible and differentiable mappings (Kobyzev et al., 2019). Hence, they can be exploited to effectively model complex probability distributions. In contrast to VAEs relying on variational approximations, flow-based models allow for latent-variable inference and likelihood evaluation in an exact and efficient manner, making themselves a perfect choice for achieving identifiability.

To this end, unifying identifiablity with flows, we propose iFlow, a framework for deep latent-variable models which allows for recovery of the true latent representations from which the observed data originates. We demonstrate that our flow-based model makes it possible to directly maximize the conditional marginal likelihood and thus achieves identifiability in a rigorous manner. We provide theoretical guarantees on the recovery of the true latent representations, and show experiments on synthetic data to validate the theoretical and practical advantages of our proposed formulation over prior approaches.

## 2 BACKGROUND

An enduring demand in statistical machine learning is to develop probabilistic models that explain the generative process that produce observations. Oftentimes, this entails estimation of density that can be arbitrarily complex. As one of the promising tools, Normalizing Flows are a family of generative models that fit such a density of exquisite complexity by pushing an initial density (base distribution) through a series of transformations. Formally, let $\mathbf{x} \in \mathcal{X} \subseteq \mathbb{R}^n$ be an observed random variable, and $\mathbf{z} \in \mathcal{Z} \subseteq \mathbb{R}^n$ a latent variable obeying a base distribution $p_Z(z)$. A normalizing flow $\mathbf{f}$ is a diffeomorphism (i.e an invertible differentiable transformation with differentiable inverse) between two topologically equivalent spaces $\mathcal{X}$ and $\mathcal{Z}$ such that $\mathbf{x} = \mathbf{f}(\mathbf{z})$. Under these conditions, the density of $\mathbf{x}$ is well-defined and can be obtained by using the change of variable formula:

$$p_X(\mathbf{x}) = p_Z(\mathbf{h}(\mathbf{x})) \left| \det\left(\frac{\partial \mathbf{h}}{\partial \mathbf{x}}\right) \right| = p_Z(\mathbf{z}) \left| \det\left(\frac{\partial \mathbf{f}}{\partial \mathbf{z}}\right) \right|^{-1}, \tag{1}$$

where $\mathbf{h}$ is the inverse of $\mathbf{f}$. To approximate an arbitrarily complex nonlinear invertible bijection, one can compose a series of such functions, since the composition of invertible functions is also invertible, and its Jacobian determinant is the product of the individual functions' Jacobian determinants. Denote $\phi$ as the diffeomorphism's learnable parameters. Optimization can proceed as follows by maximizing log-likelihood for the density estimation model:

$$\phi^* = \arg\max_{\phi} \mathbb{E}_{\mathbf{x}} \left[ \log p_Z(\mathbf{h}(\mathbf{x}; \phi)) + \log \left| \det\left(\frac{\partial \mathbf{h}(\mathbf{x}; \phi)}{\partial \mathbf{x}}\right) \right| \right]. \tag{2}$$

## 3 RELATED WORK

**Nonlinear ICA**   Nonlinear independent component analysis (ICA) is one of the biggest problems remaining unresolved in unsupervised learning. Given the observations alone, it aims to recover the inverse mixing function as well as their corresponding independent sources. In contrast with the linear case, research on nonlinear ICA is hampered by the fact that without auxiliary variables, recovering the independent latents is impossible (Hyvärinen & Pajunen, 1999). Similar impossibility result can be found in (Locatello et al., 2018). Fortunately, by exploiting additional temporal structure on the sources, recent work (Hyvarinen & Morioka, 2016; Hyvarinen et al., 2018) established the first identifiability results for deep latent-variable models. These approaches, however, do not explicitly learn the data distribution; nor are they capable of generating "fake" data.

Khemakhem et al. (2019) bridged this gap by establishing a principled connection between VAEs and an identifiable model for nonlinear ICA. Their method with an identifiable VAE (known as iVAE) approximates the true joint distribution over observed and latent variables under mild conditions. Nevertheless, due to the intractablity of KL divergence between variational approximate posterior and the true posterior, iVAE maximizes the evidence lower bound on the data log-likelihood, which in both theory and practice acts as a detriment to the achievement of identifiability.

We instead propose *identifying through flows* (normalizing flow), which maximizes the likelihood in a straightforward way, providing theoretical guarantees and practical advantages for identifiability.

**Normalizing Flows**    Normalizing Flows are a family of generative approaches that fits a data distribution by learning a bijection from observations to latent codes, and vice versa. Compared with VAEs which learn a posterior approximation to the true posterior, normalizing flows directly deal with marginal likelihood with exact inference while maintaining efficient sampling. Formally, a normalizing flow is a transform of a tractable probability distribution into a complex distribution by compositing a sequence of invertible and differentiable mappings. In practice, the challenge lies in designing a normalizing flow that satisfies the following conditions: (1) it should be bijective and thus invertible; (2) it is efficient to compute its inverse and its Jacobian determinant while ensuring sufficient capabilities.

The framework of normalizing flows was first defined in (Tabak et al., 2010) and (Tabak & Turner, 2013) and then explored for density estimation in (Rippel & Adams, 2013). Rezende & Mohamed (2015) applied normalizing flows to variational inference by introducing planar and radial flows. Since then, there had been abundant literature towards expanding this family. Kingma & Dhariwal (2018) parameterizes linear flows with the LU factorization and "$1 \times 1$" convolutions for the sake of efficient determinant calculation and invertibility of convolution operations. Despite their limits in expressive capabilities, linear flows act as essential building blocks of affine coupling flows as in (Dinh et al., 2014; 2016). Kingma et al. (2016) applied autoregressive models as a form of normalizing flows, which exhibit strong expressiveness in modelling statistical dependencies among variables. However, the forwarding operation of autoregressive models is inherently sequential, which makes it inefficient for training. Splines have also been used as building blocks of normalizing flows: Müller et al. (2018) suggested modelling a linear and quadratic spline as the integral of a univariate monotonic function for flow construction. Durkan et al. (2019a) proposed a natural extension to the framework of neural importance sampling and also suggested modelling a coupling layer as a monotonic rational-quadratic spine (Durkan et al., 2019b), which can be implemented either with a coupling architecture RQ-NSF(C) or with autoregressive architecture RQ-NSF(AR).

The expressive capabilities of normalizing flows and their theoretical guarantee of invertibility make them a natural choice for recovering the true mixing mapping from sources to observations, and thus identifiability can be rigorously achieved. In our work, we show that by aligning normalizing flows with an existing identifiability theory, it is desirable to learn an identifiable latent-variable model with theoretical guarantees of identifiability.

## 4    IDENTIFIABLE FLOW

In this section, we first introduce the identifiable latent-variable family and the theory of identifiability (Khemakhem et al., 2019) that makes it possible to recover the joint distribution between observations and latent variables. Then we derive our model, iFlow, and its optimization objective which admits principled disentanglement with theoretical guarantees of identifiability.

### 4.1    IDENTIFIABLE LATENT-VARIABLE FAMILY

The primary assumption leading to identifiability is a conditionally factorized prior distribution over the latent variables, $p_\theta(\mathbf{z}|\mathbf{u})$, where $\mathbf{u}$ is an auxiliary variable, which can be the time index in a time series, categorical label, or an additionally observed variable (Khemakhem et al., 2019).

Formally, let $\mathbf{x} \in \mathcal{X} \subseteq \mathbb{R}^n$ and $\mathbf{u} \in \mathcal{U} \subseteq \mathbb{R}^m$ be two observed random variables, and $\mathbf{z} \in \mathcal{Z} \subseteq \mathbb{R}^n$ a latent variable that is the source of $\mathbf{x}$. This implies that there can be an arbitrarily complex nonlinear mapping $\mathbf{f} : \mathcal{Z} \to \mathcal{X}$. Assuming that $\mathbf{f}$ is a bijection, it is desirable to recover its inverse by approximating using a family of invertible mappings $\mathbf{h}_\phi$ parameterized by $\phi$. The statistical

dependencies among these random variables are defined by a Bayesian net: $\mathbf{u} \to \mathbf{z} \to \mathbf{x}$, from which the following conditional generative model can be derived:

$$p(\mathbf{x}, \mathbf{z}|\mathbf{u}; \Theta) = p(\mathbf{x}|\mathbf{z}; \phi)p(\mathbf{z}|\mathbf{u}; \mathbf{T}, \boldsymbol{\lambda}), \tag{3}$$

where $p(\mathbf{x}|\mathbf{z}; \phi) \overset{\text{def}}{=} p_{\epsilon}(\mathbf{x} - \mathbf{h}^{-1}(\mathbf{z}))$ and $p(\mathbf{z}|\mathbf{u}; \mathbf{T}, \boldsymbol{\lambda})$ is assumed to be a factorized exponential family distribution conditioned upon $\mathbf{u}$. Note that this density assumption is valid in most cases, since the exponential families have universal approximation capabilities (Sriperumbudur et al., 2017). Specifically, the probability density function is given by

$$p_{\mathbf{T}, \boldsymbol{\lambda}}(\mathbf{z}|\mathbf{u}) = \prod_{i=1}^{n} p_i(\mathbf{z}_i|\mathbf{u}) = \prod_i \frac{Q_i(\mathbf{z}_i)}{Z_i(\mathbf{u})} \exp\left[\sum_{j=1}^{k} T_{i,j}(z_i)\lambda_{i,j}(\mathbf{u})\right], \tag{4}$$

where $Q_i$ is the base measure, $Z_i(\mathbf{u})$ is the normalizing constant, $T_{i,j}$'s are the components of the sufficient statistic and $\lambda_{i,j}(\mathbf{u})$ the natural parameters, critically depending on $\mathbf{u}$. Note that $k$ indicates the maximum order of statistics under consideration.

## 4.2 Identifiability Theory

The objective of identifiability is to learn a model that is subject to:

$$\text{for each quadruplet } (\Theta, \Theta', \mathbf{x}, \mathbf{z}), p_{\Theta}(\mathbf{x}) = p_{\Theta'}(\mathbf{x}) \implies p_{\Theta}(\mathbf{x}, \mathbf{z}) = p_{\Theta'}(\mathbf{x}, \mathbf{z}), \tag{5}$$

where $\Theta$ and $\Theta'$ are two different choices of model parameters that imply the same marginal density (Khemakhem et al., 2019). One possible avenue towards this objective is to introduce the definition of identifiability up to equivalence class:

**Definition 4.1. (Identifiability up to equivalence class)** Let $\sim$ be an equivalence relation on $\Theta$. A model defined by $p(\mathbf{x}, \mathbf{z}; \Theta) = p(\mathbf{x}|\mathbf{z}; \Theta)p(\mathbf{z}; \Theta)$ is said to be identifiable up to $\sim$ if

$$p_{\Theta}(\mathbf{x}) = p_{\Theta'}(\mathbf{x}) \implies \Theta \sim \Theta', \tag{6}$$

where such an equivalence relation in the identifiable latent-variable family is defined as follows:

**Proposition 4.1.** $(\phi, \tilde{\mathbf{T}}, \tilde{\boldsymbol{\lambda}})$ and $(\phi', \tilde{\mathbf{T}}', \tilde{\boldsymbol{\lambda}}')$ are of the same equivalence class if and only if there exist $\mathbf{A}$ and $\mathbf{c}$ such that $\forall\, \mathbf{x} \in \mathcal{X}$,

$$\mathbf{T}(\mathbf{h}_{\phi}(\mathbf{x})) = \mathbf{A}\mathbf{T}'(\mathbf{h}_{\phi'}(\mathbf{x})) + \mathbf{c}, \tag{7}$$

where

$$\begin{aligned}
\tilde{\mathbf{T}}(\mathbf{z}) &= (Q_1(z_1), ..., Q_n(z_n), T_{1,1}(z_1), ..., T_{n,k}(z_n)), \\
\tilde{\boldsymbol{\lambda}}(\mathbf{u}) &= (Z_1(\mathbf{u}), ..., Z_n(\mathbf{u}), \lambda_{1,1}(\mathbf{u}), ..., \lambda_{n,k}(\mathbf{u})).
\end{aligned} \tag{8}$$

One can easily verify that $\sim$ is an equivalence relation by showing its reflexivity, symmetry and transitivity. Then, the identifiability of the latent-variable family is given by Theorem 4.1 (Khemakhem et al., 2019).

**Theorem 4.1.** Let $\mathcal{Z} = \mathcal{Z}_1 \times \cdots \times \mathcal{Z}_n$ and suppose the following holds: (i) The set $\{\mathbf{x} \in \mathcal{X} | \Psi_{\epsilon}(\mathbf{x}) = 0\}$ has measure zero, where $\Psi_{\epsilon}$ is the characteristic function of the density $p_{\epsilon}$; (ii) The sufficient statistics $T_{i,j}$ in (2) are differentiable almost everywhere and $\partial T_{i,j}/\partial z \neq 0$ almost surely for $z \in \mathcal{Z}_i$ and for all $i \in \{1, ..., n\}$ and $j \in \{1, ..., k\}$. (iii) There exist $(nk+1)$ distinct priors $\mathbf{u}^0, ..., \mathbf{u}^{nk}$ such that the matrix

$$L = \begin{bmatrix} \lambda_{1,1}(\mathbf{u}^1) - \lambda_{1,1}(\mathbf{u}^0) & \cdots & \lambda_{1,1}(\mathbf{u}^{nk}) - \lambda_{1,1}(\mathbf{u}^0) \\ \vdots & \ddots & \vdots \\ \lambda_{n,k}(\mathbf{u}^1) - \lambda_{n,k}(\mathbf{u}^0) & \cdots & \lambda_{n,k}(\mathbf{u}^{nk}) - \lambda_{n,k}(\mathbf{u}^0) \end{bmatrix} \tag{9}$$

of size $nk \times nk$ is invertible. Then, the parameters $(\phi, \tilde{\mathbf{T}}, \tilde{\boldsymbol{\lambda}})$ are $\sim$-identifiable.

### 4.3 OPTIMIZATION OBJECTIVE OF iFLOW

We propose *identifying through flows* (iFlow) for recovering latent representations. Our proposed model falls into the identifiable latent-variable family with $\boldsymbol{\epsilon} = \mathbf{0}$, that is, $p_{\boldsymbol{\epsilon}}(\cdot) = \delta(\cdot)$, where $\delta$ is a point mass, i.e. Dirac measure. Note that assumption (i) in Theorem 4.1 holds true for iFlow. In stark contrast to iVAE which resorts to variational approximations and maximizes the evidence lower bound, iFlow directly maximizes the marginal likelihood conditioned on $\mathbf{u}$:

$$\max_{\Theta} p_X(\mathbf{x}|\mathbf{u};\Theta) = p_Z(\mathbf{h}_{\phi}(\mathbf{x})|\mathbf{u};\boldsymbol{\theta}) \left| \det\left( \frac{\partial \mathbf{h}_{\phi}}{\partial \mathbf{x}} \right) \right|, \tag{10}$$

where $p_Z(\cdot|\mathbf{u})$ is modeled by a factorized exponential family distribution. Therefore, the log marginal likelihood is obtained:

$$\log p_X(\mathbf{x}|\mathbf{u};\Theta) = \sum_{i=1}^{n} \left( \log Q_i(z_i) - \log Z_i(\mathbf{u}) + \mathbf{T}_i(z_i)^T \boldsymbol{\lambda}_i(\mathbf{u}) \right) + \log \left| \det\left( \frac{\partial \mathbf{h}_{\phi}}{\partial \mathbf{x}} \right) \right|, \tag{11}$$

where $z_i$ is the $i$th component of the source $\mathbf{z} = \mathbf{h}_{\phi}(\mathbf{x})$, and $\mathbf{T}$ and $\boldsymbol{\lambda}$ are both $n$-by-$k$ matrices. Here, $\mathbf{h}_{\phi}$ is a normalizing flow of any kind. For the sake of simplicity, we set $Q_i(z_i) = 1$ for all $i$'s and consider maximum order of sufficient statistics of $z_i$'s up to 2, that is, $k = 2$. Hence, $\mathbf{T}$ and $\boldsymbol{\lambda}$ are given by

$$\mathbf{T}(\mathbf{z}) = \begin{pmatrix} z_1^2 & z_1 \\ z_2^2 & z_2 \\ \vdots & \vdots \\ z_n^2 & z_n \end{pmatrix} \quad \text{and} \quad \boldsymbol{\lambda}(\mathbf{u}) = \begin{pmatrix} \xi_1 & \eta_1 \\ \xi_2 & \eta_2 \\ \vdots & \vdots \\ \xi_n & \eta_n \end{pmatrix}. \tag{12}$$

Therefore, the optimization objective is to minimize

$$\mathcal{L}(\Theta) = \mathbb{E}_{(\mathbf{x},\mathbf{u})\sim p_D} \left[ \left( \sum_{i=1}^{n} \log Z_i(\mathbf{u}) \right) - \text{trace}\left( \mathbf{T}(\mathbf{z})\boldsymbol{\lambda}(\mathbf{u})^T \right) - \log \left| \det\left( \frac{\partial \mathbf{h}_{\phi}}{\partial \mathbf{x}} \right) \right| \right], \tag{13}$$

where $p_D$ denotes the empirical distribution, and the first term in (13) is given by

$$
\begin{aligned}
\sum_{i=1}^{n} \log Z_i(\mathbf{u}) &= \log \int_{\mathbb{R}^n} \left( \prod_{i=1}^{n} Q_i(z_i) \right) \exp\left( \text{trace}\left( \mathbf{T}(\mathbf{z})\boldsymbol{\lambda}(\mathbf{u})^T \right) \right) d\mathbf{z} \\
&= \log \int_{\mathbb{R}^n} \exp\left( \sum_{i=1}^{n} \xi_i z_i^2 + \eta_i z_i \right) d\mathbf{z} \\
&= \log \prod_{i=1}^{n} \int_{\mathbb{R}} \exp\left( \xi_i z_i^2 + \eta_i z_i \right) dz_i \\
&= \log \prod_{i=1}^{n} \left( \sqrt{-\frac{\pi}{\xi_i}} \right) \exp\left( -\frac{\eta_i^2}{4\xi_i} \right) \\
&= \sum_{i=1}^{n} \left( \log \sqrt{-\frac{\pi}{\xi_i}} - \frac{\eta_i^2}{4\xi_i} \right).
\end{aligned}
\tag{14}
$$

In practice, $\boldsymbol{\lambda}(\mathbf{u})$ can be parameterized by a multi-layer perceptron with learnable parameters $\boldsymbol{\theta}$, where $\boldsymbol{\lambda}_{\boldsymbol{\theta}} : \mathbb{R}^m \to \mathbb{R}^{2n}$. Here, $m$ is the dimension of the space in which $\mathbf{u}$'s lies. Note that $\xi_i$ should be strictly negative in order for the exponential family's probability density function to be finite. Negative softplus nonlinearity can be exploited to force this constraint. Therefore, optimization proceeds by minimizing the following closed-form objective:

$$\min_{\Theta} \mathcal{L}(\Theta) = \mathbb{E}_{(\mathbf{x},\mathbf{u})\sim p_D} \left[ \sum_{i=1}^{n} \left( \log \sqrt{-\frac{\pi}{\xi_i}} - \frac{\eta_i^2}{4\xi_i} \right) - \text{trace}\left( \mathbf{T}(\mathbf{z})\boldsymbol{\lambda}_{\boldsymbol{\theta}}(\mathbf{u})^T \right) - \log \left| \det\left( \frac{\partial \mathbf{h}_{\phi}}{\partial \mathbf{x}} \right) \right| \right]. \tag{15}$$

where $\Theta = \{\boldsymbol{\theta}, \boldsymbol{\phi}\}$.

### 4.4 IDENTIFIABILITY OF iFLOW

The identifiability of our proposed model, iFlow, is characterized by Theorem 4.2.

**Theorem 4.2.** Minimizing $\mathcal{L}_\Theta$ with respect to $\Theta$, in the limit of infinite data, learns a model that is $\sim$-identifiable.

*Proof.* Minimizing $\mathcal{L}_\Theta$ with respect to $\Theta$ is equivalent to maximizing the log conditional likelihood, $\log p_X(\mathbf{x}|\mathbf{u}; \Theta)$. Given infinite amount of data, maximizing $\log p_X(\mathbf{x}|\mathbf{u}; \Theta)$ will give us the true marginal likelihood conditioned on $\mathbf{u}$, that is, $p_X(\mathbf{x}|\mathbf{u}; \hat{\Theta}) = p_X(\mathbf{x}|\mathbf{u}; \Theta^*)$, where $\hat{\Theta} = \arg\max_\Theta \log p_X(\mathbf{x}|\mathbf{u}; \Theta)$ and $\Theta^*$ is the true parameter. According to Theorem 4.1, we obtain that $\hat{\Theta}$ and $\Theta^*$ are of the same equivalence class defined by $\sim$. Thus, according to Definition 4.1, the joint distribution parameterized by $\Theta$ is identifiable up to $\sim$. $\qquad\square$

Consequently, Theorem 4.2 guarantees strong identifiability of our proposed generative model, iFlow. Note that unlike Theorem 3 in (Khemakhem et al., 2019), Theorem 4.2 makes no assumption that the family of approximate posterior distributions contains the true posterior. And we show in experiments that this assumption is unlikely to hold true empirically.

## 5 SIMULATIONS

To evaluate our method, we run simulations on a synthetic dataset. This section will elaborate on the details of the generated dataset, implementation, evaluation metric and fair comparison with existing methods.

### 5.1 DATASET

We generate a synthetic dataset where the sources are non-stationary Gaussian time-series, as described in (Khemakhem et al., 2019): the sources are divided into $M$ segments of $L$ samples each. The auxiliary variable $\mathbf{u}$ is set to be the segment index. For each segment, the conditional prior distribution is chosen from the exponential family (4), where $k = 2$, $Q_i(z_i) = 1$, and $T_{i,1}(z_i) = z_i^2$, $T_{i,2}(z_i) = z_i$, and the true $\lambda_{i,j}$'s are randomly and independently generated across the segments and the components such that their variances obey a uniform distribution on $[0.5, 3]$. The sources to recover are mixed by an invertible multi-layer perceptron (MLP) whose weight matrices are ensured to be full rank.

### 5.2 IMPLEMENTATION DETAILS

The mapping $\boldsymbol{\lambda}_\theta$ that outputs the natural parameters of the conditional factorized exponential family distribution is parameterized by a multi-layer perceptron with the activation of the last layer being the softplus nonlinearity. Additionally, a negative activation is taken on the second-order natural parameters in order to ensure its finiteness. The bijection $\mathbf{h}_\phi$ is modeled by RQ-NSF(AR) (Durkan et al., 2019b) with the flow length of 10 and the bin 8, which gives rise to sufficient flexibility and expressiveness. For each training iteration, we use a mini-batch of size 64, and an Adam optimizer with learning rate chosen in $\{0.01, 0.001\}$ to optimize the learning objective (15).

### 5.3 EVALUATION METRIC

As a standard measure used in ICA, the mean correlation coefficient (MCC) between the original sources and the corresponding predicted latents is chosen to be the evaluation metric. A high MCC indicates the strong correlation between the identified latents recovered and the true sources. In experiments, we found that such a metric can be sensitive to the synthetic data generated by different random seeds. We argue that unless one specifies the overall generating procedure including random seeds in particular any comparison remains debatable. This is crucially important since most of the existing works failed to do so. Therefore, we run each simulation of different methods through seed 1 to seed 100 and report averaged MCCs with standard deviations, which makes the comparison fair and meaningful.

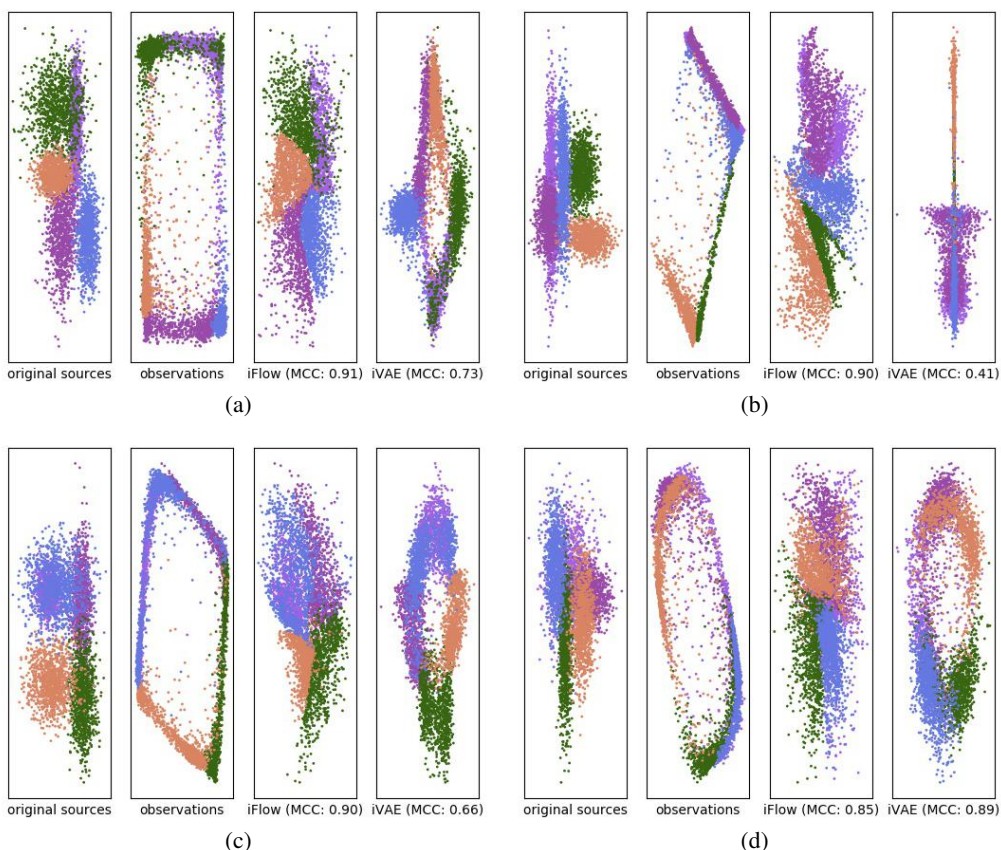

Figure 1: Visualization of 2-D cases (best viewed in color).

## 5.4 COMPARISON AND RESULTS

We compare our model, iFlow, with iVAE. These two models are trained on the same synthetic dataset aforementioned, with $M = 40$, $L = 1000$, $n = d = 5$. For visualization, we also apply another setting with $M = 40$, $L = 1000$, $n = d = 2$. To evaluate iVAE's identifying performance, we use the original implementation that is officially released with exactly the same settings as described in (Khemakhem et al., 2019) (cf. Appendix A.2).

First, we demonstrate a visualization of identifiablity of these two models in a 2-D case ($n = d = 2$) as illustrated in Figure 1, in which we plot the original sources (latent), observations and the identified sources recovered by iFlow and iVAE, respectively. Segments are marked with different colors. Clearly, iFlow outperforms iVAE in identifying the original sources while preserving the original geometry of source manifold. It is evident that the latents recovered by iFlow bears much higher resemblance to the true latent sources than those by iVAE in the presence of some trivial indeterminacies of scaling, global sign and permutation of the original sources, which are inevitable even in some cases of linear ICA. This exhibits consistency with the definition of identifiability up to equivalence class that allows for existence of an affine transformation between sufficient statistics, as described in Proposition 4.1. As shown in Figure 1(a), 1(c), and 1(d), iVAE achieves inferior identifying performance in the sense that its estimated latents tend to retain the manifold of the observations. Notably, we also find that despite the relatively high MCC performance of iVAE in Figure 1(d), iFlow is much more likely to recover the true geometric manifold in which the latent sources lie. In Figure 1(b), iVAE's recovered latents collapses in face of a highly nonlinearly mixing case, while iFlow still works well in identifying the true sources. Note that these are not rare occurrences. More visualization examples can be found in Appendix A.3.

---

https://github.com/ilkhem/iVAE/

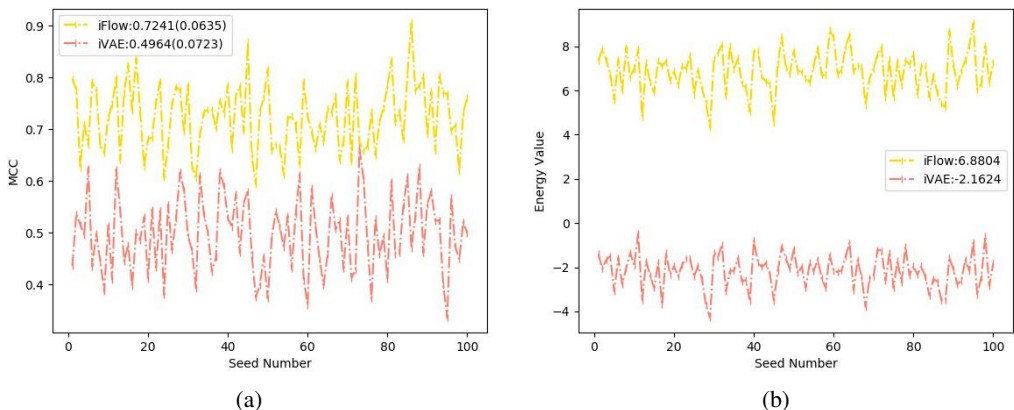

Figure 2: Comparison of identifying performance (MCC) and the energy value (likelihood in logarithm) versus seed number, respectively.

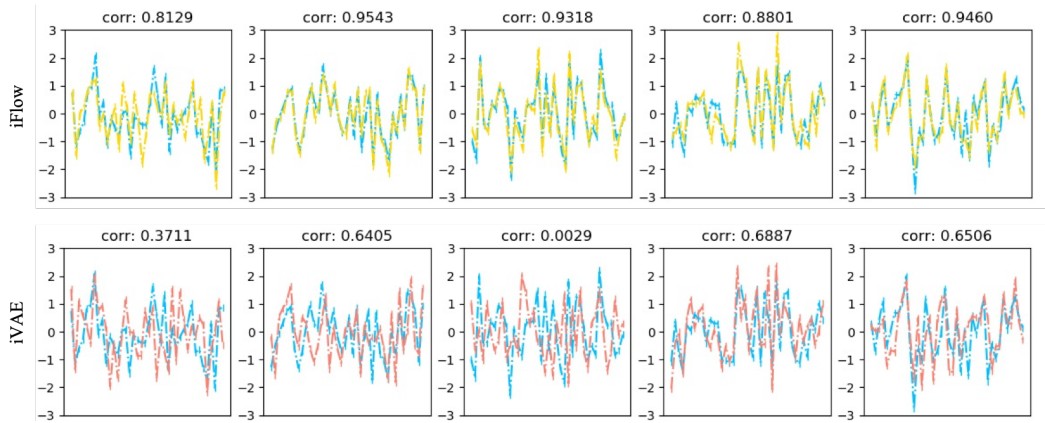

Figure 3: Comparison of identifying performance (correlation coefficient) in each single dimension of the latent space, respectively. The dashed cyan line represents the source signal.

Second, regarding quantitative results as shown in Figure 2(a), our model, iFlow, consistently outperforms iVAE in MCC by a considerable margin across different random seeds under consideration while experiencing less uncertainty (standard deviation as indicated in the brackets). Moreover, Figure 2(b) also showcases that the energy value of iFlow is much higher than that of iVAE, which serves as evidence that the optimization of the evidence lower bound, as in iVAE, would lead to suboptimal identifiability. As is borne out empirically, the gap between the evidence lower bound and the conditional marginal likelihood is inevitably far from being negligible in practice. For clearer analysis, we also report the correlation coefficients for each source-latent pair in each dimension. As shown in Figure 3, iFlow exhibits much stronger correlation than does iVAE in each single dimension of the latent space.

Finally, we investigate the impact of different choices of activation for generating natural parameters of the exponential family distribution (see Appendix A.1 for details). Note that all of these choices are valid since theoretically the natural parameters form a convex space. However, iFlow(Softplus) achieves the highest identifying performance, suggesting that the range of softplus allows for greater flexibility, which makes itself a good choice for natural parameter nonlinearity.

# 6 CONCLUSIONS

Among the most significant goals of unsupervised learning is to learn the disentangled representations of observed data, or to identify original latent codes that generate observations (i.e. identifiability). Bridging the theoretical and practical gap of rigorous identifiability, we have proposed iFlow, which directly maximizes the marginal likelihood conditioned on auxiliary variables, establishing a natural framework for recovering original independent sources. In theory, our contribution provides a rigorous way to achieve identifiability and hence the recovery of the joint distribution between observed and latent variables that leads to principled disentanglement. Extensive empirical studies confirm our theory and showcase its practical advantages over previous methods.

ACKNOWLEDGMENTS

We thank Xiaoyi Yin for helpful initial discussions.
**This work is in loving memory of *Kobe Bryant* (1978-2020) ...**

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

## A APPENDIX

### A.1 ABLATION STUDY ON ACTIVATIONS FOR NATURAL PARAMETERS

Figure 4 demonstrates the comparison of MCC of iFlows implemented with different nonlinear activations for natural parameters and that of iVAE, in which relu+eps denotes the ReLU activation added by a small value (e.g. 1e-5) and sigmoid×5 denotes the Sigmoid activation multiplied by 5.

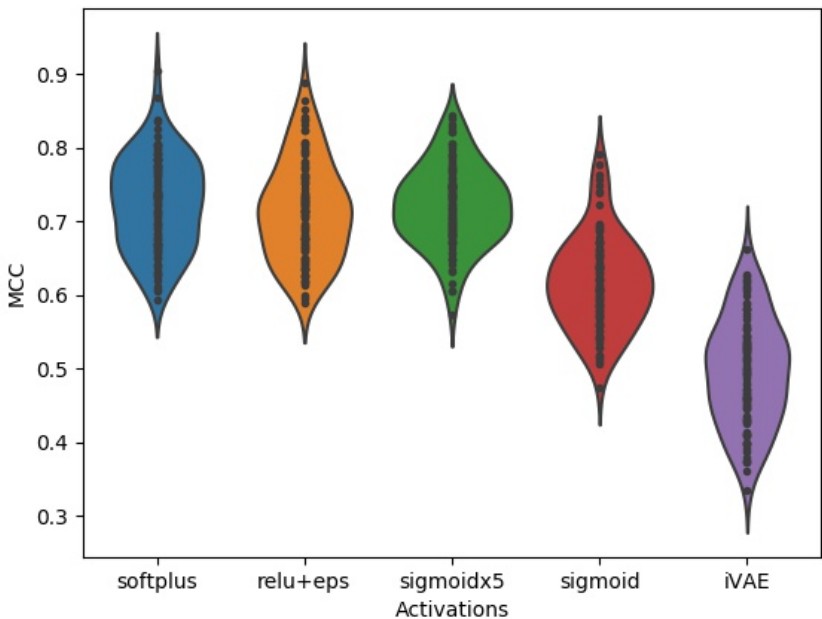

Figure 4: Comparison of MCC of iFlows implemented with different nonlinear activations for natural parameters and that of iVAE (best viewed in color).

### A.2 IMPLEMENTATION DETAILS OF IVAE

As stated in Section 5.4, to evaluate iVAE's identifying performance, we use the original implementation that is officially released with the same settings as described in (Khemakhem et al., 2019). Specifically, in terms of hyperparameters of iVAE, the functional parameters of the decoder and the inference model, as well as the conditional prior are parameterized by MLPs, where the dimension of the hidden layers is chosen from {50, 100, 200}, the activation function is a leaky RELU or a leaky hyperbolic tangent, and the number of layers is chosen from {3, 4, 5, 6}. Here we report all averaged MCC scores of different implementations for iVAE as shown in Table 1.

Table 1 indicates that adding more layers or more hidden neurons does not improve MCC score, precluding the possibility that expressive capability is not the culprit of iVAE inferior performance. Instead, we argue that the assumption (i) of Theorem 3 in (Khemakhem et al., 2019) (i.e the family of approximate posterior distributions contains the true posterior) often fails or is hard to satisfy in practice, which is one of the major reasons for the inferior performance of iVAE. Additionally, Figure 2(b) demonstrates that the energy value of iFlow is much higher than that of iVAE, which provides evidence that optimizing the evidence lower bound, as in iVAE, leads to suboptimal identifiability.

Table 1: Averaged MCC scores versus different iVAE implementations

| NUM_HIDDENS | NUM_LAYERS | AVG MCC |
|---|---|---|
| 50 | 3 | 0.4964(±0.0723) |
| 50 | 4 | 0.4782(±0.0740) |
| 50 | 5 | 0.4521(±0.0719) |
| 50 | 6 | 0.4276(±0.0686) |
| 100 | 3 | 0.4965(±0.0693) |
| 100 | 4 | 0.4696(±0.0685) |
| 100 | 5 | 0.4555(±0.0736) |
| 100 | 6 | 0.4253(±0.0795) |
| 200 | 3 | 0.4961(±0.0753) |
| 200 | 4 | 0.4723(±0.0707) |
| 200 | 5 | 0.4501(±0.0686) |
| 200 | 6 | 0.4148(±0.0703) |

## A.3 VISUALIZATION OF 2D CASES

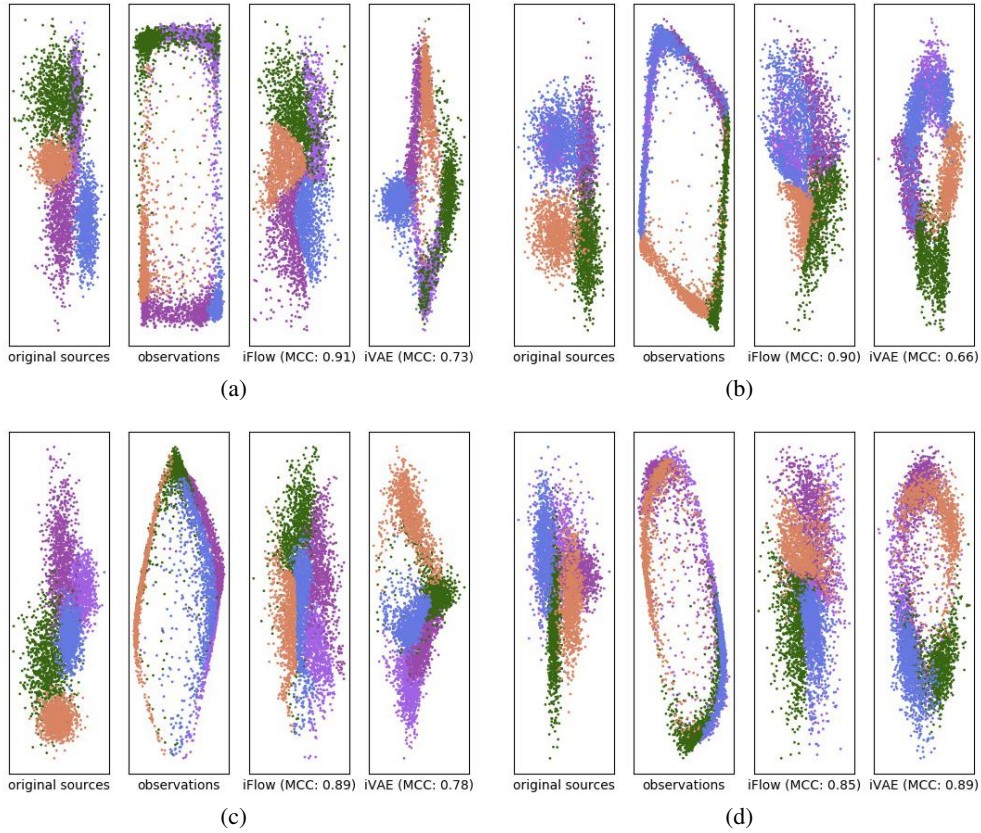

Figure 5: Visualization of 2-D cases (i) (best viewed in color).

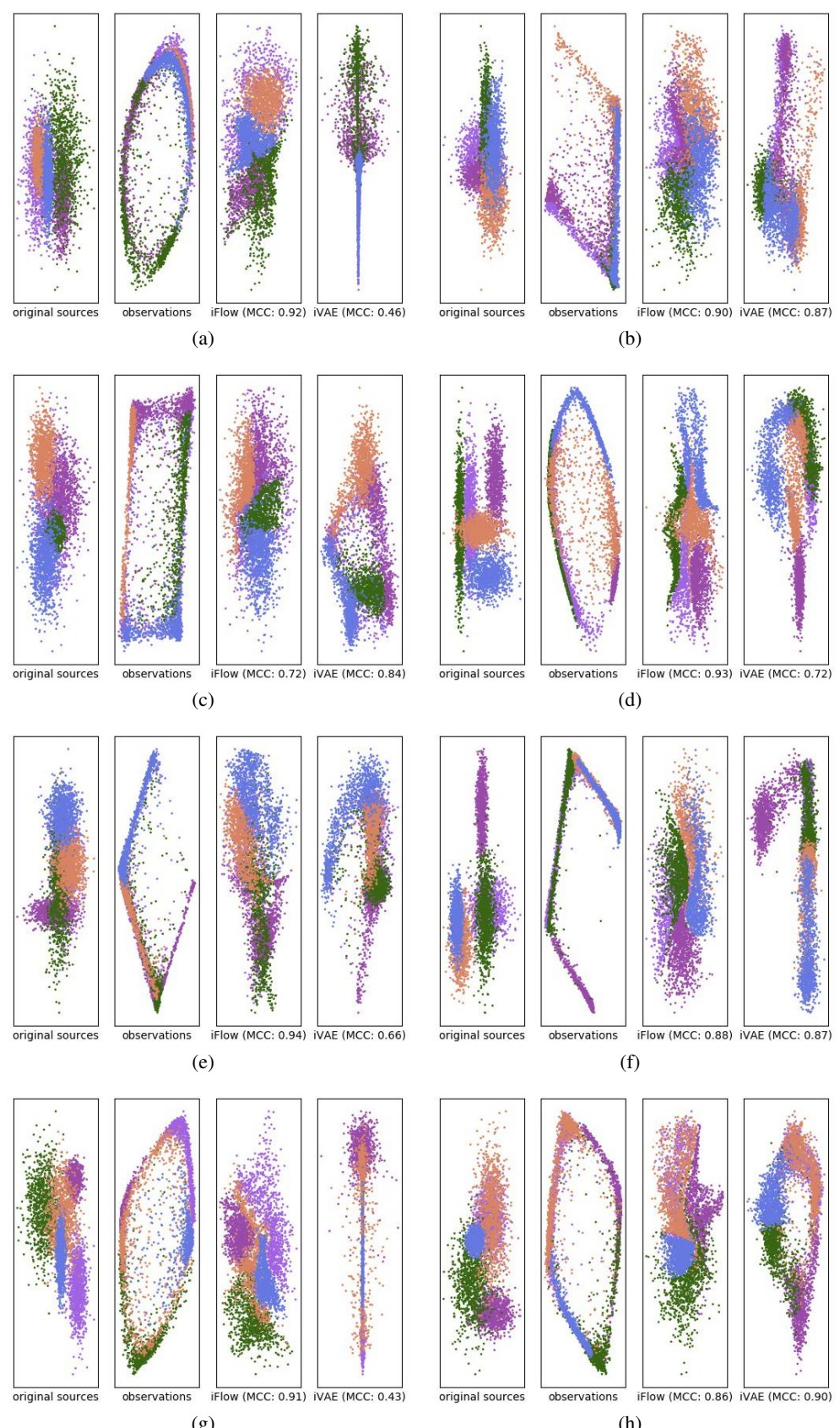

Figure 6: Visualization of 2-D cases (ii) (best viewed in color).

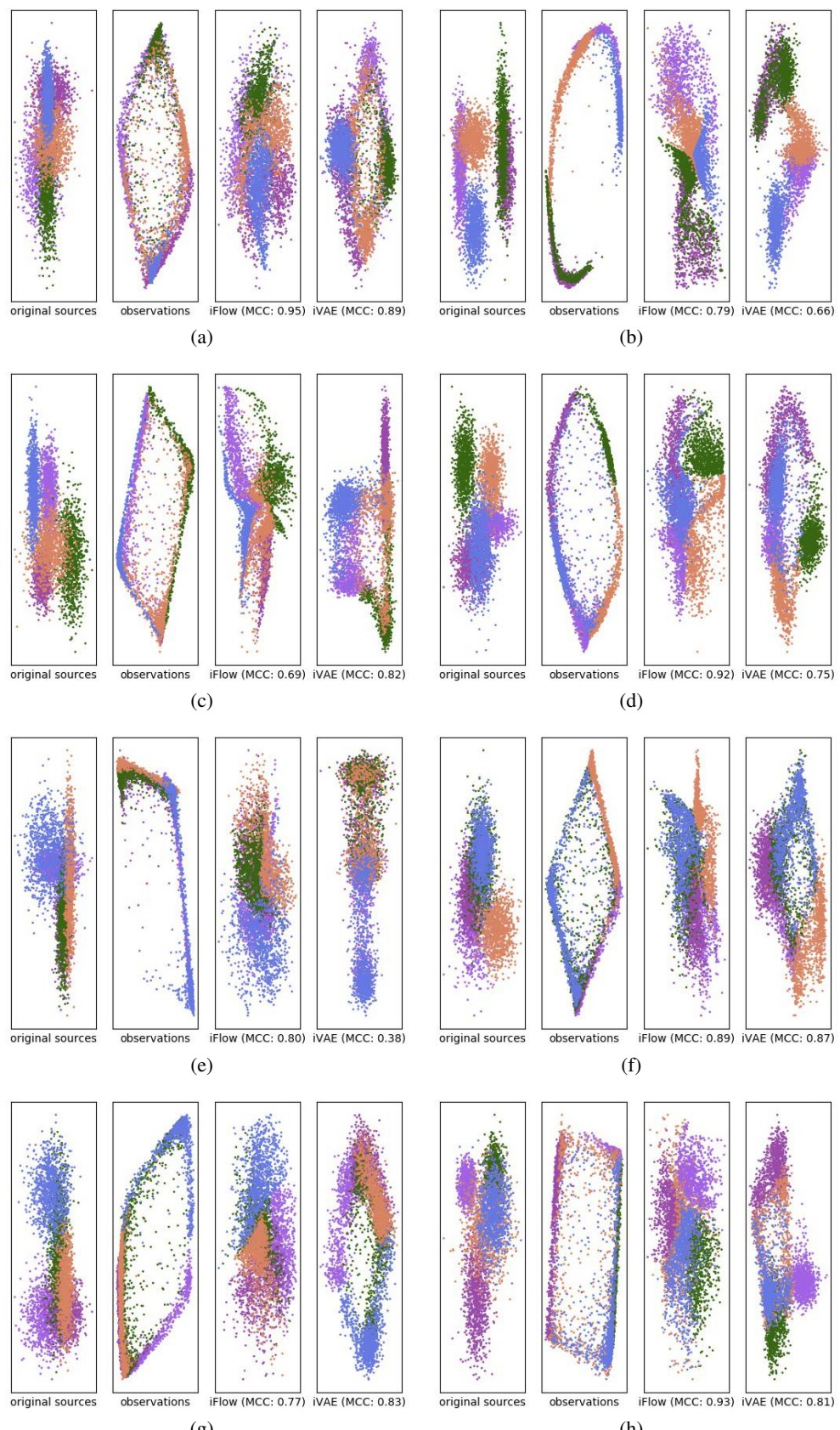

Figure 7: Visualization of 2-D cases (iii) (best viewed in color).

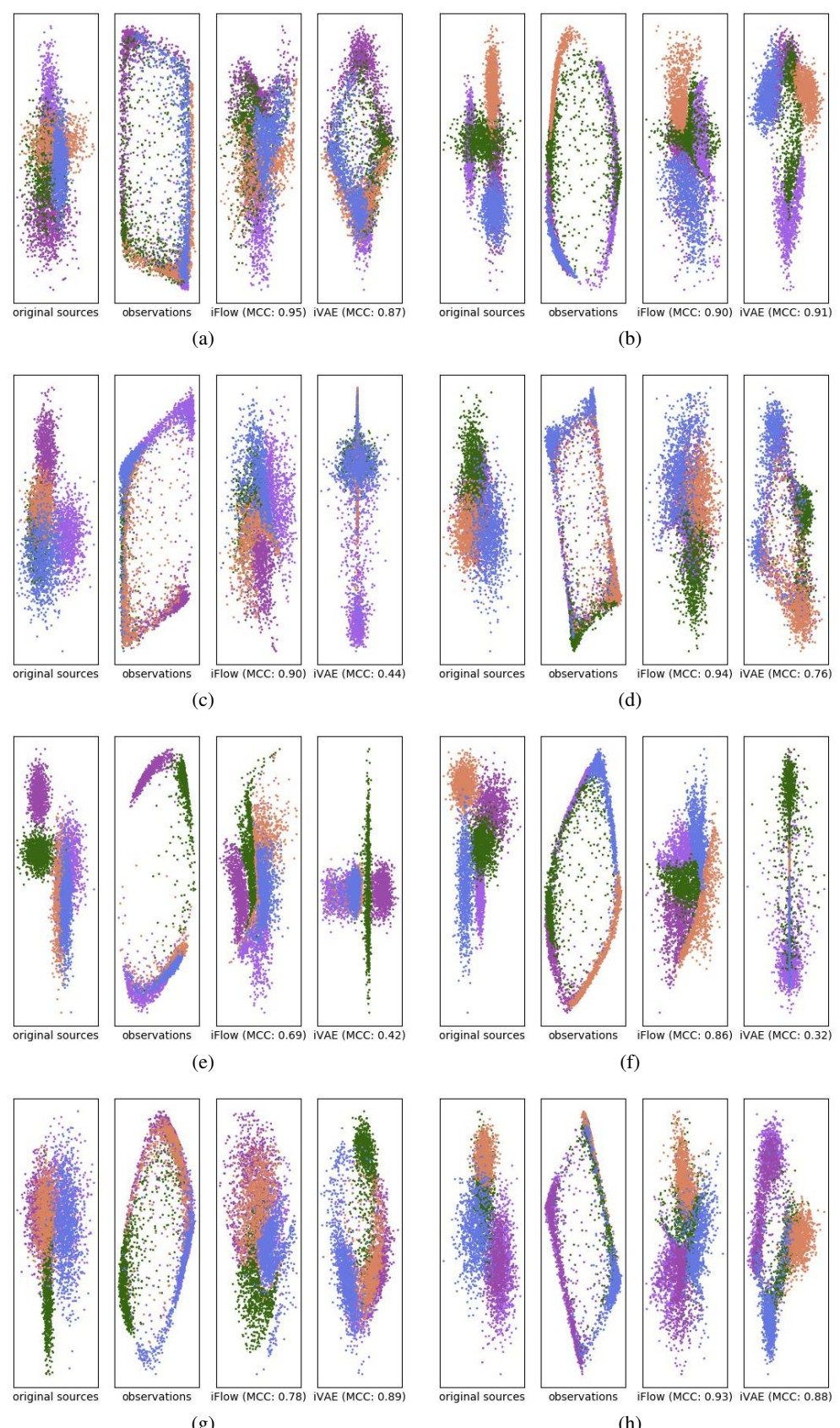

Figure 8: Visualization of 2-D cases (iv) (best viewed in color).

