# OpenReview forum: "Identifying through Flows for Recovering Latent Representations"
_ICLR.cc/2020/Conference — Accept (Poster)_

### Official Review · AnonReviewer1 · 2019-10-25
**Official Blind Review #1**

**Rating:** 6

**Review:**

This paper is about learning an identifiable generative model, iFlow, that builds upon a recent result on nonlinear ICA. The key idea is providing side information to identify the latent representation, i.e., essentially a prior conditioned on extra information such as labels and restricting the mapping to flows for being able to compute the likelihood. As the loglikelihood of a flow model is readily available, a direct approach can be used for learning that optimizes both the prior and the observation model.

The paper is very clear and very easy to follow. The idea is quite clear, and the direct approach is really attractive. Unfortunately, the experimental section is quite limited and does not fully study representational aspects. There is only an illustrative simulation on synthetic data, that in a sense verifies the theory. I was also not able to see the additional insight that the identifiability theory in 4.2 provides additional to Khemakhem et al. (2019). Please clarify.

My main concern is that this model is actually a supervised model that learns a mapping from u to x via z. Hence, the theoretical ‘identifiability’ guarantee needs to be stated with some care as this depends on the choice of an arbitrary u. For example if we set x = u, will learning take place? Please comment.

Overall, I like the approach but I am uncertain about the level of novelty. For such a paper, one should expect a much more involved computational study. In this respect, I feel that the paper could be accepted but it certainly feels as if it needs more computational results as otherwise the original contribution would be too incremental for ICLR standards. I am giving a provisional reject in the hope that the authors will provide convincing arguments about their original contributions for clarification.


**Experience Assessment:**

I have read many papers in this area.

**Review Assessment: Checking Correctness Of Derivations And Theory:**

I assessed the sensibility of the derivations and theory.

**Review Assessment: Checking Correctness Of Experiments:**

I assessed the sensibility of the experiments.

**Review Assessment: Thoroughness In Paper Reading:**

I read the paper at least twice and used my best judgement in assessing the paper.

---

> ### Author Response · Authors · 2019-11-08
> **Response to Reviewer #1**
>
> Thank you for your time and valuable comments on our paper. We will use these suggestions to make improvements to the paper. Here are our responses to the main questions.
>
>
> Q1: Provide convincing arguments about our original contributions for clarification.
>
> Response to Q1:
>
> Identifiability has been a longstanding goal of research in unsupervised representation learning. The identifiability theory and the iVAE method were established in Khemakhem et al. (2019). In our work, we observed that due to the intractability of KL divergence between variational approximate posterior and the true posterior, iVAE has to maximize ELBO that often leads to suboptimal solutions empirically (as is confirmed in Figure 1, 2, 3 in our paper) which also fails to provide theoretical guarantees of model identifiability (since the assumption (i) of Theorem 3 as proposed in Khemakhem et al. (2019) does not hold true often in practice).
>
> To bridge this gap both in theory and in practice, our original contribution in this paper is to propose an identifiable framework for estimating latent sources using a flow-based model (iFlow) based upon the identifiability theory. This allows for direct maximization of the marginal conditional likelihood, which leads to better identifiability results in contrast to iVAE (as is shown in Figure 2(a) and also in Figure 1, 3 and Appendix B).
> Specifically, in terms of contributions, we derive the optimization objective in analytical form, making it possible to train iFlow in an end-to-end manner. Furthermore, we prove theoretical guarantees on the identifiability of iFlow, while relaxing the assumption (i) of Theorem 3 as proposed in Khemakhem et al. (2019). Our contribution also includes an extensive empirical evaluation of iFlow, which supports our theory and formulation. Experimental results demonstrate that iFlow indeed outperforms iVAE with a considerable margin.
>
> In a nutshell, our contributions include (1) our flow-based model for achieving identifiability; (2) proving theoretical guarantees of model identifiability; and (3) clear empirical advantages over prior methods.
>
>
> Q2: The experimental section is quite limited and does not fully study representational aspects. There is only an illustrative simulation on synthetic data, that in a sense verifies the theory.
>
> Response to Q2:
>
> Running simulations on synthetic data is for empirical verification of non-linear ICA methods (including ours). The ambition of non-linear ICA methods is to recover the true latent unknown sources that generate observations. However, as pointed out in [6], true latent variables of real data are unavailable or not well-defined. Therefore, evaluating non-linear ICA methods can be problematic and ambiguous. Even though one can run such a method on a real dataset, there is no feasible way to quantitatively evaluate the method, since the true latent sources remain inaccessible in the real world. Hence, the state-of-the-art [1, 3] focuses on the synthetic scenario where true latent sources happen to be independent by construction for synthetic data. Besides, note that Figure 2 (a) shows the comparison of MCC (a standard measure used in ICA) between iFlow and iVAE, indicating that iFlow consistently outperforms iVAE by a considerable margin. This empirical result verifies our theory: directly optimizing the likelihood with flows instead of ELBO achieves better identifiability with theoretical guarantees.
>
>
> Q3: I was also not able to see the additional insight that the identifiability theory in 4.2 provides additional to Khemakhem et al. (2019). Please clarify.
>
> Response to Q3:
>
> The identifiability theory introduced in 4.2 was first established in [1]. We cited it for clarity before our approach (iFlow) is proposed in 4.3. Thanks for pointing it out. We will rephrase our description for greater clarity and add citation in Section 4.2.
>
> [TO BE CONTINUED]
>
>
> REFERENCES
> [1] Ilyes Khemakhem, Diederik P Kingma, and Aapo Hyvarinen. Variational autoencoders and nonlinear ica: A unifying framework. arXiv preprint arXiv:1907.04809, 2019.
> [2] Aapo Hyvarinen and Hiroshi Morioka. Unsupervised feature extraction by time-contrastive learning and nonlinear ica. In Advances in Neural Information Processing Systems, pp. 3765–3773, 2016.
> [3] Aapo Hyvarinen, Hiroaki Sasaki, and Richard E Turner. Nonlinear ica using auxiliary variables and generalized contrastive learning. arXiv preprint arXiv:1805.08651, 2018.
> [4] Aapo Hyvarinen and Petteri Pajunen. Nonlinear independent component analysis: Existence and ¨ uniqueness results. Neural Networks, 12(3):429–439, 1999.
> [5] Francesco Locatello, Stefan Bauer, Mario Lucic, Sylvain Gelly, Bernhard Scholkopf, and Olivier ¨ Bachem. Challenging common assumptions in the unsupervised learning of disentangled representations. arXiv preprint arXiv:1811.12359, 2018.
> [6] Emile Mathieu, Tom Rainforth, N. Siddharth, and Yee Whye Teh.   Disentangling disentanglement in variational autoencoders, 2018.

---

> ### Author Response · Authors · 2019-11-08
> **Response to Reviewer #1 (Continued)**
>
> Q4: My main concern is that this model is actually a supervised model that learns a mapping from u to x via z. Hence, the theoretical ‘identifiability’ guarantee needs to be stated with some care as this depends on the choice of an arbitrary u. For example if we set x = u, will learning take place? Please comment.
>
> Response to Q4:
>
> Thanks for the excellent point and the question. We will rephrase our statements about the identifiability guarantee to clarify the role of the auxiliary variable $u$. Here we provide context on the introduction of $u$ in our setting:
>
> The problem we consider in this paper is actually nonlinear ICA problem, whose definition is to assume that the observed data $x = (x_1, …, x_n)$ is generated by an arbitrary (but smooth and invertible) transformation $f$ of the latent variables $z = (z_1, …, z_n)$ as $x=f(z)$. The goal is then to recover the inverse function $f^{-1}$ as well as the independent components $z_i$ based on observations of x alone [3].
>
> However, research in nonlinear ICA has been hampered by the fact that such simple approaches to nonlinear ICA are not identifiable, in stark contrast to the linear ICA case. Hyvarinen and Pajunen [4] gave a theoretical proof that in the nonlinear ICA case, if there is no temporal or similar structure in the data, any model is unidentifiable. Moreover, Locatello et al. [5] also provided the theorem of impossibility result showing that unsupervised disentanglement learning is fundamentally impossible for arbitrary generative models. All of these prior works indicate that without any auxiliary information, it is theoretically impossible to develop an identifiable model.
>
> Therefore, state-of-the-art works [2, 3, 1] resorted to auxiliary data, based on the idea that the independent components are dependent on some additional auxiliary variable, while being conditionally mutually independent given the auxiliary variable. Note that by incorporating additional information, the resulting model proves to be identifiable in theory [2, 3, 1].
>
> In our paper, we follow this paradigm, where $x$ denotes the observations that are generated from the latent unknown sources $z$ to be recovered, and $u$ denotes the auxiliary variable other than $x$. As we pointed out in our paper, $u$ can be time index in a time series, categorical label, or any other kind of additionally observed variable.
>
>
> REFERENCES
> [1] Ilyes Khemakhem, Diederik P Kingma, and Aapo Hyvarinen. Variational autoencoders and nonlinear ica: A unifying framework. arXiv preprint arXiv:1907.04809, 2019.
> [2] Aapo Hyvarinen and Hiroshi Morioka. Unsupervised feature extraction by time-contrastive learning and nonlinear ica. In Advances in Neural Information Processing Systems, pp. 3765–3773, 2016.
> [3] Aapo Hyvarinen, Hiroaki Sasaki, and Richard E Turner. Nonlinear ica using auxiliary variables and generalized contrastive learning. arXiv preprint arXiv:1805.08651, 2018.
> [4] Aapo Hyvarinen and Petteri Pajunen. Nonlinear independent component analysis: Existence and ¨ uniqueness results. Neural Networks, 12(3):429–439, 1999.
> [5] Francesco Locatello, Stefan Bauer, Mario Lucic, Sylvain Gelly, Bernhard Scholkopf, and Olivier ¨ Bachem. Challenging common assumptions in the unsupervised learning of disentangled representations. arXiv preprint arXiv:1811.12359, 2018.
> [6] Emile Mathieu, Tom Rainforth, N. Siddharth, and Yee Whye Teh.   Disentangling disentanglement in variational autoencoders, 2018.

---

### Official Review · AnonReviewer2 · 2019-10-27
**Official Blind Review #2**

**Rating:** 6

**Review:**

## Overview
The paper tackles the identifiability problem in generative modeling, i.e., recovering the true latent representations from which the observed data originates. The paper argues that identifiable variational autoencoder (iVAE) suffers from intractability issue which leads to suboptimal solutions. The paper instead proposes an identifiable normalizing flows (iFlow) method as an alternative. The proposed iFlow outperforms iVAE in experiments using synthetic data. The paper is very well motivated and well supports its claim.

## Summary of the contributions
1. The paper proposes iFlow, an identifiable normalizing flow method which allows recovery of true latent space from observed data.
2. The paper shows iFlow outperforms iVAE in synthetic experiments.
3. The paper provides theoretical justification on the identifiability of the proposed iFlow method.

## Overall feedback
I find the paper well written and well organized. Easy to follow the content even though I am not expert on this matter. The paper provides both theoretical justification and empirical experimental validation which show the superior performance of proposed iFlow method. So I am leaning towards accepting the paper. The theory seems correct but I did not check all the equations and proofs in depth so I am not very confident about my rating.

## Suggestions
1. Please make sure all equations are properly punctuated.
2. The comparison with iVAE seems tricky since they use different types of architectures. Can you also report the hyper parameters used for iVAE?

**Experience Assessment:**

I do not know much about this area.

**Review Assessment: Checking Correctness Of Derivations And Theory:**

I assessed the sensibility of the derivations and theory.

**Review Assessment: Checking Correctness Of Experiments:**

I assessed the sensibility of the experiments.

**Review Assessment: Thoroughness In Paper Reading:**

I made a quick assessment of this paper.

---

> ### Author Response · Authors · 2019-11-08
> **Response to Reviewer #2**
>
> Thank you for your time and valuable comments on our paper. We will use these suggestions to make improvements to the paper. Here are our responses to the main questions.
>
> Q1: Please make sure all equations are properly punctuated.
>
> Response to Q1:
>
> Thanks for the suggestion. We will make sure all equations are properly punctuated in the latest version (e.g Eqn. (5) should end with a period, and (1-4) should end with commas).
>
>
> Q2: The comparison with iVAE seems tricky since they use different types of architectures. Can you also report the hyper parameters used for iVAE?
>
> Response to Q2:
>
> Thanks for pointing this out. As stated in Section 5.4 of our paper, to evaluate iVAE’s identifying performance, we use the original implementation that is officially released with the same settings as described in [1]. Specifically, in terms of hyperparameters of iVAE, the functional parameters of the decoder and the inference model, as well as the conditional prior are chosen to be MLPs, where the dimension of the hidden layers is chosen in {50, 100, 200}, the activation function is a leaky RELU or a leaky hyperbolic tangent, and the number of layers is chosen in {3, 4, 5, 6}. We tried different types of architectures with the above hyperparameters and reported the best MCC score for iVAE.
> Here we report all MCC scores of different implementations for iVAE:
>
> NUM_HIDDENS                 NUM_LAYERS                    AVG MCC
> --------------------------------------------------------------------------------------
>           50                                        3                       $0.4964(\pm 0.0723)$
>           50                                        4                       $0.4782(\pm 0.0740)$
>           50                                        5                       $0.4521(\pm 0.0719)$
>           50                                        6                       $0.4276(\pm 0.0686)$
>          100                                       3                       $0.4965(\pm 0.0693)$
>          100                                       4                       $0.4696(\pm 0.0685)$
>          100                                       5                       $0.4555(\pm 0.0736)$
>          100                                       6                       $0.4253(\pm 0.0795)$
>          200                                       3                       $0.4961(\pm 0.0753)$
>          200                                       4                       $0.4723(\pm 0.0707)$
>          200                                       5                       $0.4501(\pm 0.0686)$
>          200                                       6                       $0.4148(\pm 0.0703)$
>
> This indicates that adding more layers or more hidden neurons does not improve MCC score, precluding the possibility that expressive capability is not the culprit. Therefore, the comparison between iFlow and iVAE in our paper is fair. We argue that the assumption (i) of Theorem 3 in [1] (i.e the family of approximate posterior distributions contains the true posterior) often fails or is hard to satisfy in practice, which is one of the major reasons for the inferior performance of iVAE. Additionally, Figure 2(b) in our paper demonstrates that the energy value of iFlow is much higher than that of iVAE, which provides evidence that optimizing the evidence lower bound, as in iVAE, leads to suboptimal identifiability.
>
> In contrast, we propose an identifiable framework using normalizing flow, which directly optimizes the training objective (i.e the marginal likelihood conditioned on u) instead of its lower bound. We showed the theoretical guarantees on the identifiability of iFlow by relaxing the assumption (i) of Theorem 3 in [1]. We derived the training objective in analytical form, making it possible to train iFlow in an end-to-end manner. The extensive experimental results and ablation study empirically confirm our theory.
>
>
> REFERENCES
> [1] Ilyes Khemakhem, Diederik P Kingma, and Aapo Hyvarinen. Variational autoencoders and nonlinear ica: A unifying framework. arXiv preprint arXiv:1907.04809, 2019.
> [2] Aapo Hyvarinen and Hiroshi Morioka. Unsupervised feature extraction by time-contrastive learning and nonlinear ica. In Advances in Neural Information Processing Systems, pp. 3765–3773, 2016.
> [3] Aapo Hyvarinen, Hiroaki Sasaki, and Richard E Turner. Nonlinear ica using auxiliary variables and generalized contrastive learning. arXiv preprint arXiv:1805.08651, 2018.
> [4] Aapo Hyvarinen and Petteri Pajunen. Nonlinear independent component analysis: Existence and ¨ uniqueness results. Neural Networks, 12(3):429–439, 1999.
> [5] Francesco Locatello, Stefan Bauer, Mario Lucic, Sylvain Gelly, Bernhard Scholkopf, and Olivier ¨ Bachem. Challenging common assumptions in the unsupervised learning of disentangled representations. arXiv preprint arXiv:1811.12359, 2018.
> [6] Emile Mathieu, Tom Rainforth, N. Siddharth, and Yee Whye Teh.   Disentangling disentanglement in variational autoencoders, 2018.

---

### Decision · Program_Chairs · 2019-12-19

**Decision:**

Accept (Poster)

**Comment:**

Main content:

Blind review #1 summarizes it well:

This paper is about learning an identifiable generative model, iFlow, that builds upon a recent result on nonlinear ICA. The key idea is providing side information to identify the latent representation, i.e., essentially a prior conditioned on extra information such as labels and restricting the mapping to flows for being able to compute the likelihood. As the loglikelihood of a flow model is readily available, a direct approach can be used for learning that optimizes both the prior and the observation model.

--

Discussion:

Reviewer questions were mostly about clarification, which the authors addressed during the rebuttal period.

--

Recommendation and justification:

All reviewers agree the paper is a weak accept based on degree of depth, novelty, and impact.